# Bayesian Maximum-A-Posteriori Approach with Global and Local Regularization to Image Reconstruction Problem in Medical Emission Tomography

**DOI:** 10.3390/e21111108

**Published:** 2019-11-12

**Authors:** Natalya Denisova

**Affiliations:** Institute of Theoretical and Applied Mechanics, Novosibirsk 630090, Russia; NVDenisova2011@mail.ru

**Keywords:** image reconstruction, Bayesian Maximum a Posteriori approach, entropy prior probability, global statistical regularization, local statistical regularization, PET, SPECT

## Abstract

The Bayesian approach Maximum a Posteriori (MAP) provides a common basis for developing statistical methods for solving ill-posed image reconstruction problems. MAP solutions are dependent on a priori model. Approaches developed in literature are based on prior models that describe the properties of the expected image rather than the properties of the studied object. In this paper, such models have been analyzed and it is shown that they lead to global regularization of the solution. Prior models that are based on the properties of the object under study are developed and conditions for local and global regularization are obtained. A new reconstruction algorithm has been developed based on the method of local statistical regularization. Algorithms with global and local regularization were compared in numerical simulations. The simulations were performed close to the real oncologic single photon emission computer tomography (SPECT) study. It is shown that the approach with local regularization produces more accurate images of ‘hot spots’, which is especially important to tumor diagnostics in nuclear oncology.

## 1. Introduction

Image reconstruction belongs to the class of ill-posed inverse problems of mathematical physics [1]. The method of solution for this class of problems was developed in 1963 by A.Tikhonov and called regularization [2]. Regularization introduces a priori information regarding the problem to obtain well-behaved inverse. The introduction of a priori information entails the necessity of choosing an unknown parameter, which is called the regularization parameter. In Tikhonov’s approach, this parameter plays a role of a weight factor, which controls the competition between a priori information and the measured data. Initially, the method was developed in the form of ‘global regularization’, in which a single parameter controls the solution. However, it was found that global regularization often provides too smoothed solutions, even in the early practical applications. The idea of ‘local regularization’ was suggested to locally regulate ‘a level of smoothness’ to improve the method [3]. The reconstruction algorithm based on local regularization was developed and applied in the x-ray tomographic systems ‘CPT-1000′ and ‘CPT-5000′ [4]. In both, numerical simulations and clinical practice, it was proven that local regularization produces images that are deemed of higher quality in comparison to global regularization. In [2,3], a priori information was introduced in deterministic form.

In 1967, the physicist V. Turchin suggested using the Bayesian method of Maximum a Posteriori (MAP) for solving inverse ill-posed problems with stochastic data, naming this approach ‘statistical regularization’ [5,6]. The statistical regularization method introduces a priori information in the probabilistic form. Later, in the 70–80s the Bayesian method for solving image restoration and image reconstruction problems had become popular. The main difficulty of the Bayesian approach is the determination of a priori probability density function. Two forms of prior probability are used for solving the problems of restoring and reconstruction of images: entropy concept and Gibbs probability distribution. Jaynes [7,8,9] suggested prior information, based on the entropy concept, for solving problems with limited, but noise-free data. This approach was called the maximum entropy (ME) method. In 1972, the ME approach was applied by Frieden to solving the image restoration problem [10]. After a few years, the ME approach was successfully applied in the restoration of radio astronomy images in the paper by Gull and Daniell [11]. In [10,11], the image restoration problem was solved as a constrained optimization problem. J. Skilling developed the approach based on the Bayesian method Maximum a Posteriori with entropy-based prior probability (MAP-ENT) [12]. In 1979, Minerbo used the ME approach for the tomography problems [13]. The tomographic task was solved as constrained optimization problem. The probabilistic approach MAP-ENT for solving tomographic problems was developed and applied to plasma tomography by W. von der Linden [14]. In [15], the relation between MAP-ENT and ME was studied. An improvement in reconstructed image quality by the MAP-ENT algorithm over the ME was demonstrated. In [16], the MAP-ENT approach was developed and applied for nuclear medicine.

Besag [17] and D. Geman and S. Geman [18] theoretically justified another form of a priori probability known as ‘the Gibbs prior’. The approach that was based on the Bayesian method Maximum a Posteriori with Gibbs a priori probability (MAP-GIBBS) is widely studied for applications in nuclear medicine. Geman and McClare first applied this approach to nuclear medicine in 1985 [19].

In the present paper, both of these prior models are studied in the context of their application to image reconstruction problems in nuclear medicine, namely, positron emission tomography (PET) and single photon emission computer tomography (SPECT). Currently, non-regularized Maximum Likelihood-based (ML) algorithms are used for image reconstruction in PET and SPECT. A fast version of the ML algorithm, which is known as Ordered Subset Expectation Maximization (OSEM), is applied in most existing PET and SPECT clinical systems. Images that were obtained using the OSEM are often noisy; therefore, the post-smoothing procedure is applied that leads to over-smoothing of fine structures. Huge efforts are being made in literature to develop regularized algorithms that are based on the Bayesian MAP approach. It was expected that the regularized MAP algorithms should provide more accurate reconstruction of fine structures in comparison to the non-regularized OSEM. However, rather minor differences between OSEM and MAP-GIBBS images were found [20]. Additionally, in [21], the MAP-ENT algorithm did not result in image improvement in comparison to the OSEM. We suggest the hypothesis that the lack of improvement in reconstruction of fine structures is associated with global regularization in the existing MAP algorithms. Global regularization smoothes the solution equally in the entire area, regardless of the local characteristics of a source function and, therefore, fine structures may be over-smoothed or lost. The aim of this study is to develop the Bayesian MAP approach with local regularization and to check the suggested hypothesis. Previously, the idea of local regularization for statistical Bayesian MAP approach has not been considered in the literature.

## 2. Theory

### 2.1. Bayesian Approach for Solving Image Reconstruction Problems in Nuclear Medicine

In SPECT and PET diagnostic procedures, a patient is administered a radiopharmaceutical that is distributed in various organs of the patient body. Let $n=\{n_{j},j=1\ldots J\}$ represent the radiopharmaceutical concentration distribution in the three-dimensional (3D) object. The object is divided into J voxels. The number of gamma photons produced through radioactive decay is denoted by $f=\{f_{j},j=1\ldots J\}$. f is a random vector with mean $\bar{f}=\{{\bar{f}}_{j},j=1\ldots J\}$, which is assumed to be proportional to radiopharmaceutical concentration:(1)$${\bar{f}}_{j}\propto n_{j}$$

The photons that are emitted by the radiopharmaceutical are collected in a system of detectors located around the patient. Measurements are denoted by the random vector $g=\{g_{i},i=1\ldots I\}$. The data acquisition process is modeled by the system of linear equations:(2)$$\sum_{j}a_{ij}f_{j}=g_{i}$$

$g_{i}$ are the measured data in the i-th detector cell, $a_{ij}$ is the random operator that describes which fraction of photons emitted from the voxel j is detected in the detector pixel i. In Figure 1, an example of measured clinical data is demonstrated. A left anterior oblique projection that was obtained in myocardial perfusion SPECT imaging is shown. One can see that these data have stochastic properties. In PET and SPECT, the Poisson distribution gives the likelihood function:(3)$$P(g|\bar{f})=\prod_{i}e^{-{\bar{g}}_{i}}\frac{{\bar{g}}_{i}{}^{g_{i}}}{g_{i}!}$$
with means:(4)$${\bar{g}}_{i}=\sum_{j}^{}{\bar{a}}_{ij}{\bar{f}}_{j}$$

The matrix element ${\bar{a}}_{ij}$ stands for the probability that a photon emitted from the voxel j will be detected in the detector pixel i. Having stochastic data g, the reconstruction problem is formulated as a classical problem of mathematical statistics: what is the probability density $P(\bar{f}|g)$ for the solution $\bar{f}$ given g? The probability $P(\bar{f}|g)$ is determined according to the Bayesian method Maximum *a Posteriori* (MAP):(5)$$P(\bar{f}|g)=\frac{P(\bar{f})P(g|\bar{f})}{\int P(\bar{f})P(g|\bar{f})d\bar{f}}$$

$P(\bar{f})$ is the a priori probability density function and $P(g|\bar{f})$ is the likelihood distribution of the observed data. MAP estimation (in logarithmic form) provides the solution of the reconstruction problem:(6)$$\tilde{\bar{f}}=\arg_{\bar{f}>0}\max\{\ln P(\bar{f})+\ln P(g|\bar{f})\}$$

The main difficulty of the Bayesian approach is the determination of the a priori probability $P(\bar{f})$. There are two different classes of imaging problems: restoration of distorted images and tomographic reconstruction of objects. When solving the image restoration problems, it is usually impossible to include a priori information about real objects. A priori information refers to the expected image to be restored. In contrast, when reconstructing tomographic images, one can often propose the physical model of the object to be reconstructed and determine a priori information by using this model. Currently, in tomographic problems, one uses a priori information, which was initially developed to solve the image restoration problems. In this paper, we discuss these approaches to specifying a priori probability in both the image restoration problem and the image reconstruction problem. We consider the three most widely used forms of a priori information: no prior, Gibbs prior, and entropy-based prior.

### 2.2. Maximum-Likelihood-Based Image Reconstruction Method (ML)

When a researcher does not have any prior information, the simple way is to assume that all possibilities are equally probable and Bayesian estimation (6) reduces to the Maximum Likelihood (ML) solution:(7)$$\tilde{\bar{f}}=\arg_{\bar{f}}\max\{\ln P(g|\bar{f})\}$$

The log-likelihood function is written:(8)$$\ln P(g|\bar{f})=\sum_{i}[g_{i}\ln\sum_{j}a_{ij}{\bar{f}}_{j}-\sum_{j}a_{ij}{\bar{f}}_{j}-\ln g_{j}!]$$

In (8), and further in the formulas, by $a_{ij}$ is meant ${\bar{a}}_{ij}$. The ML method gives the following resulting solution (ML algorithm) [22]:(9)$${\tilde{\bar{f}}}_{j}^{n+1}=\frac{{\tilde{\bar{f}}}_{j}^{n}}{\sum_{i}a_{ij}}\sum_{i}\frac{g_{i}a_{ij}}{\sum_{j}a_{ij}{\tilde{\bar{f}}}_{j}^{n}}$$

${\tilde{\bar{f}}}_{j}^{n}$ is an estimation of ${\bar{f}}_{j}$ on the n-th iteration step. As stated in the Introduction, a fast version of the ML algorithm (9), known as the OSEM, is applied in most existing PET and SPECT clinical systems.

### 2.3. Bayesian Image Reconstruction Method MAP-GIBBS

#### 2.3.1. Gibbs Prior Based on Image Properties

Initially, the prior probability in the form of Gibbs distribution was developing for restoration of degraded images [18]. The image was considered as pixel/voxel gray levels in a lattice-like physical system. It was assumed that the Markov Random Field (MRF) model reflects the expectations of smoothness and captures the distributions of ‘units of grey’. According to the Hammersly–Clifford theorem, the MRF has the distribution, which is described by the Gibbs probability [23]:(10)$$P(\bar{f})=\frac{1}{Z}\exp(-\beta U(\bar{f}))$$

$U(\bar{f})$ is called the energy function, Z is a normalizing constant, and β is a positive constant. The energy function $U(\bar{f})$ is designed, so that the expected image configurations are those for which neighboring pixels have similar intensities. Still, sharp gradients can occur. This local constraint is written as:(11)$$U(\bar{f})=\sum_{j}\sum_{k\in c_{j}}w_{jk}V_{}({\bar{f}}_{j}-{\bar{f}}_{k})$$
where $c_{j}$ denotes the set of neighbors of pixel *j*, V is a potential function defined on pairwise cliques of neighboring pixels, $w_{jk}$ is a weight factor specified the closeness of pixels *j* and *k*. The prior probability (10) in logarithmic form is written as:(12)$$\ln P(\bar{f})=-\beta \sum_{j}\sum_{k\in c_{j}}w_{jk}V({\bar{f}}_{j}-{\bar{f}}_{k})$$

In [8], this form of prior probability was adopted to solve the problem of image reconstruction in SPECT with the following argumentation: (1) isotope concentration being fairly constant within small regions of common tissue type; neighboring pixels are more likely than not to have similar isotope concentrations; (2) there are may be a sharp gradient in the intensity values of neighboring pixels: sharp boundaries will occur between two tissue types, or between two organs. Taking into account the log-likelihood function (8) and prior probability (12), the Bayesian estimation (6) is written as:(13)$$\tilde{\bar{f}}=\underset{\vec{f}\geq 0}{\arg}\max\{-\beta \sum_{j}\sum_{k\in c_{j}}w_{jk}V_{}({\bar{f}}_{j}-{\bar{f}}_{k})+\sum_{i}[g_{i}\ln\sum_{j}a_{ij}{\bar{f}}_{j}-\sum_{j}a_{ij}{\bar{f}}_{j}-\ln g_{j}!]\}$$

The MAP method with Gibbs a priori distribution (MAP-GIBBS) gives the resulting solution [24]:(14)$${\tilde{\bar{f}}}_{j}^{n+1}=\frac{{\tilde{\bar{f}}}_{j}^{n}}{(\sum_{i}a_{ij}+\beta \frac{\partial U(\bar{f}){|}_{\bar{f}\in c_{j}}}{\partial {\bar{f}}_{j}^{}})}\sum_{i}\frac{g_{i}a_{ij}}{\sum_{j}a_{ij}{\tilde{\bar{f}}}_{j}^{n}}$$

This algorithm is widely studied in the literature for solving PET and SPECT reconstruction problems. A variety of energy functions $U(\bar{f})$ have been proposed in the literature. They differ by the choice of potential function V that assigns cost to differences between neighboring pixels/voxels. As was noted above, on one side, piecewise smoothness is preferred, but, on the other side, a priori information should be tolerant for large differences between neighboring voxels. For example, edge-preserving functions V were developed in [24,25]:(15)$$V_{}(|{\bar{f}}_{j}-{\bar{f}}_{k}|)=\frac{{\left({\bar{f}}_{j}-{\bar{f}}_{k}\right)}^{2}}{2\delta^{2}+{\left({\bar{f}}_{j}-{\bar{f}}_{k}\right)}^{2}}$$
(16)$$V_{}(|{\bar{f}}_{j}-{\bar{f}}_{k}|)=\left\{ \begin{array}{l} \frac{1}{2\delta^{2}}{\left({\bar{f}}_{j}-{\bar{f}}_{k}\right)}^{2},\;\mathrm{if}\left|{\bar{f}}_{j}-{\bar{f}}_{k}\right|\leq \delta \\ \{\left|{\bar{f}}_{j}-{\bar{f}}_{k}\right|-\frac{\delta^{}}{2}\}/\delta^{},\;\mathrm{else} \end{array}\right.$$

The parameter δ in (15) and (16) controls the difference between ${\bar{f}}_{j}$ and ${\bar{f}}_{k}$ in neighboring voxels. The following comments can be added analyzing the MRF-based Gibbs a priori probability (12). The MRF-based prior model was developed for image restoration, segmentation, and texture modeling, and, formally, it reflects image properties rather than the properties of a real physical object. Expressions (15) and (16) are looking as mathematical constructs, but it is not very clear a physical nature of V. The parameter β is a constant and it leads to global regularization in the MAP-GIBBS algorithm (14).

#### 2.3.2. Gibbs a Priori Probability Based on a Closed System Model

In Section 2.3.1, we discussed Gibbs prior probability that reflects the properties of an expected image. In the present section, the Gibbs prior probability is related to the object to be reconstructed. In nuclear medicine, the real object to be reconstructed is a steady-state (during data acquisition time) spatial distribution of radiopharmaceutical particles in a patient body. The particles do not interact with each other due to low radionuclide concentration. We consider this distribution as a closed physical system, placed in thermostat (patient body) with a certain temperature T. It is assumed that some force field causes non-uniform spatial particles distribution. We do not consider the specific fields and forces that hold the particles. Under the influence of the field, the different parts of the system are in different conditions, so the system might be spatially non-uniform and, at the same time, equilibrium [26]. The canonical Gibbs distribution describes such a system:(17)$$P(n)=\frac{1}{Z}\exp[-\frac{H(n)}{kT}]$$
where H(n) is the Hamiltonian of the system of *N* particles. For simplicity, we assume that the volume of each voxel is unit and the concentration $n_{j}$ determines the number of particles in the voxel j. The Hamiltonian is written for the non-interacting particles:(18)$$H(n)=\sum_{j=1}^{J}\sum_{i=1}^{n_{j}}\left\{\frac{p_{i}^{2}}{2m}+U(q_{j})\right\}$$
where q and p are the coordinates and momenta of particles. According to the Gibbs theorem, an average energy is a constant value. For neighboring voxels *k* and *j* this can be written as:(19)$$<H>=\frac{3}{2}kT_{k}+U_{k}+\mu_{k}=\frac{3}{2}kT_{j}+U_{j}+\mu_{j}=const$$
where μ is a chemical potential per particle (without field), U is a potential energy of particles in the field. Taking into account that in thermostat *T* = *const* one obtains from (19):(20)dU=−dμ

On the other side, in thermodynamically equilibrium system with *T* = *const*
(21)dμ=vdp
where dp is the pressure difference between neighboring voxels and $v=1/n_{j}$ is the specific volume of one particle [26]. The potential U is defined from (20) and (21) as a function of pressure gradient. In the case of non-interacting particles, U can be represented as the particle concentration gradient in a discrete form, as follows:(22)$$U_{j}=\frac{1}{n_{j}}\sum_{k\in c_{j}}w_{jk}V(n_{k}-n_{j})$$

$w_{jk}$ is a weight factor specified the closeness of pixels *j* and *k*, $c_{j}$ denotes the set of neighbors of pixel *j*. Substituting (18) into (17) and integrating the Gibbs distribution (17) on momenta one obtains:(23)$$P(n)=K\exp[-\frac{\sum_{j=1}^{J}\sum_{i=1}^{n_{j}}U_{j}}{kT}]$$

K is a positive constant that is a result of multiplying $\frac{1}{Z}$ and the constant of integration on momenta. The probability density (23) can be written in logarithmic form as:(24)$$\ln P(n)=-\frac{1}{kT}\sum_{j}\sum_{k\in c_{j}}w_{jk}V(n_{k}-n_{j})+const$$

While taking into account (1), the prior probability (24) can be rewritten (up to a constant):(25)$$\ln P(\bar{f})=-\beta \sum_{j}\sum_{k\in c_{j}}w_{jk}V({\bar{f}}_{k}-{\bar{f}}_{j})$$

When comparing expressions (12) and (25), one can see that they are the same. This approach, which is based on the statistical physics model, allow for us to answer the comments at the end of the previous Section 2.3.1. Different forms of the potential function (15), (16) correspond to different state equations of the system. The parameter β must be constant throughout the solution area. Therefore, Gibbs a priori probability necessarily leads to global regularization in the MAP-GIBBS algorithm.

### 2.4. Bayesian Image Reconstruction Method MAP-ENT

Another *a priori* model, which is based on the entropy principle, was successfully applied in the fields of X-ray-, radio- and gamma-astronomy, plasma tomography [11,12,13,14,15], but less in medicine tomography. Applied to solve the problem of astronomical image restoring, the Maximum Entropy approach has demonstrated that resolved and unresolved sources can be restored with reliability [27,28]. These results are of interest for nuclear medicine in the context of the similarity between astronomical images and ‘hot spots’ tumor images. In the absence of correlations, the entropy-based method is superior to the Gibbs’ approach in ‘hot spots’ identifying. Therefore, in the present paper, this method is considered from the point of view of its potential in nuclear medicine.

#### 2.4.1. Entropy a Priori Probability Based on Image Properties

The ME approach that is based on image properties can be formulated, as follows. An image is considered as a set of boxes in which a large number of ‘particles’ are distributed. Different authors make different sense for the term ‘particles’, as ‘radiance units’, luminance quanta, units of gray, silver particles. According to the combinatorial theorem, the number of different ways of filling the boxes is given by.
(26)$$W=\frac{N_{0}!}{\prod_{j}N_{j}!}$$

j is a box index, $N_{j}$ is the number of ‘radiance units’ in the *j*-th box, and $N_{0}$ is the total number of ‘radiance units’. While using the Stirling approximation, one can write *W* in the logarithmic form, as follows:(27)$$\ln W=-N_{0}\sum_{j=1}^{J}(N_{j}/N_{0})\ln(N_{j}/N_{0})$$

Boltzmann has found the connection between entropy *S* and multiplicity *W*:(28)S=klnW

*k* is a coefficient depending on the taken dimension. The distribution with higher entropy might be realized in the greatest number of ways and, hence, it yields the most probable solution consistent with given data. Therefore, entropy can be used as prior probability density in the Bayesian approach Maximum a Posteriori:(29)$$\ln P(N_{j}|N_{0})=\ln W$$

One usually uses the following form for the entropy-based prior:(30)$$P(\bar{f})=-\beta \sum_{j}{\bar{f}}_{j}\ln{\bar{f}}_{j}$$
where ${\bar{f}}_{j}$ is an expected value of unknown source function, β is a constant associated with the relation between $N_{j}/N_{0}$ and ${\bar{f}}_{j}$, which controls the competition between entropy-based prior information and measured data. In [12], the entropy-based prior was developed in the form:(31)$$P(\bar{f})=\sum_{j}({\bar{f}}_{j}-m_{j}-{\bar{f}}_{j}\ln\frac{{\bar{f}}_{j}}{m_{j}})$$

$m_{j}$ is the default model. In our next work, we are going to study and compare the forms (30) and (31) in application to nuclear medicine. In the present work, entropy form (30) was used. In nuclear medicine, the Bayesian approach with entropy prior was developed for images reconstruction in SPECT [16]. Substitution of the entropy prior (30) and condition probability (8) into (6) gives the Maximum a Posteriori estimation with entropy-based prior (MAP-ENT):(32)$$\tilde{\bar{f}}=\underset{\vec{f}\geq 0}{\arg}\max\{-\beta \sum_{j=1}^{J}{\bar{f}}_{j}\ln{\bar{f}}_{j}+\sum_{i}g_{i}\ln\sum_{j}a_{ij}{\bar{f}}_{j}-\sum_{j}a_{ij}{\bar{f}}_{j}-\ln g_{j}!\}$$

The iteration relation for determining $\tilde{\bar{f}}$ was obtained in [16], as:(33)$${\bar{f}}_{j}^{n+1}=\exp(-1+\gamma (\sum_{i}\frac{g_{i}a_{ij}}{\sum_{j}a_{ij}{\bar{f}}_{j}^{n}}-\sum_{i}a_{ij}))$$
where n is an iteration number. The sum $\sum_{i}$ is taken over the set of observations. The iteration solution (33) depends on the regularization parameter γ=1/β. In our calculations, the multiplicative form of the MAP-ENT algorithm with ${\bar{f}}_{j}^{0}=1/e$ was used:(34)$${\bar{f}}_{j}^{n+1}={\bar{f}}_{j}^{n}\exp(\gamma (\sum_{i}\frac{g_{i}a_{ij}}{\sum_{j}a_{ij}{\bar{f}}_{j}^{n}}-\sum_{i}a_{ij}))$$

Analyzing entropy prior (30), we can add the following comments: (a) derivation of Formulas (26)–(30) was based on the properties of an image; (b) the role of the parameter γ=1/β in (33) is not very clear, it is also not clear whether it can be locally changed.

#### 2.4.2. Entropy Prior Based on the Boltzmann Isolated System Model

The spatially non-uniform distribution of non-interacting radiopharmaceutical particles in the patient’s body can be described while using the model of the non-equilibrium Boltzmann ideal gas. In contrast to the prior model (30), here we consider the distribution of particles in phase space. Up to a constant factor, the Boltzmann entropy is written as:(35)S=−∫n(p,q)lnn(p,q)dpdq
where n(p,q) is the particle density in the phase space. Replace the variables
(36)$$dpdq=dq2\pi {\left(2m\right)}^{3/2}\varepsilon^{1/2}d\varepsilon$$
where ε is particle energy. Entropy functional (35) can be rewritten as:(37)$$S=-\sum_{j}\{\int n(q_{j},\varepsilon_{})\ln n(q_{j},\varepsilon_{})2\pi {\left(2m\right)}^{3/2}\varepsilon^{1/2}d\varepsilon +const\}$$

In (37), integration over the spatial coordinate *dq* is replaced by summation over discrete voxels $q_{j}$. We assume that the voxel size is so small, so that an equilibrium energy distribution is established in each voxel $q_{j}$ and the particle energy distribution function ρ(ε) has a very sharp maximum at $\varepsilon (q_{j})=\bar{\varepsilon }(q_{j})$, where $\bar{\varepsilon }$ is a mean energy. According to the Boltzmann theory for isolated systems, the mean energy value should be the same in the entire system $\bar{\varepsilon }(q_{j})=\varepsilon_{0}$. The entropy functional can be rewritten as:(38)$$S=-\alpha \sum_{j}n(q_{j},\varepsilon_{0})\ln n(q_{j},\varepsilon_{0})\varepsilon_{0}{}^{1/2}\Delta \varepsilon +const.$$

The ‘width’ Δε is defined from normalization $\int \rho (\varepsilon )d\varepsilon =\rho (\varepsilon_{0})\Delta \varepsilon =1$. As a result, the entropy functional (38) is presented in its standard form:(39)$$S=-\frac{\alpha }{\rho (\varepsilon_{0})}\sum_{j}n(q_{j})\ln n(q_{j})+const,$$
where $n(q_{j})$ is the radionuclide particles concentration in the voxel with co-ordinate $q_{j}$. Taking (1) into account, (39) is written as:(40)$$S=-\beta \sum_{j}{\bar{f}}_{j}\ln{\bar{f}}_{j}+const$$

When comparing expressions (31) and (40), one can see that they are the same. The regularization parameter β depends on the particles mean energy. β is a constant throughout the solution area.

#### 2.4.3. Entropy Prior Based on Open System Model

Open systems are the systems that are exchanged with environment by energy, matter, and information. Therefore, emitting objects are open systems. In SPECT and PET diagnostic procedures, a radiopharmaceutical is injected into a patient body. Different organs have different metabolism rate for the injected radiopharmaceutical. In the same organ, healthy and ill tissues can also have different metabolism rate. The accumulative model is considered, which describes a final steady-state non-equilibrium spatial distribution of radiopharmaceutical in a patient body. Due to radioactive decay, radiopharmaceutical emits gamma photons, so a patient body can be considered as an emitting open system. It is assumed that the processes of nuclear excitation and de-excitation occur at the same rate during all the time of the patient examination procedure. In an open system, there can be regions in which the average energy values will be different. The total entropy is equal to the sum of the entropies of the subsystems since these regions are independent. For the case of two selected regions *C1* and *C2* with mean energies $\varepsilon_{0}$ and $\varepsilon_{1}$, the expression (38) is changed, as follows:(41)$$S=-\frac{\alpha }{\rho (\varepsilon_{0})}\sum_{j\in C1}n(q_{j})\ln n(q_{j})+\frac{\alpha }{\rho (\varepsilon_{1})}\sum_{j\in C2}n(q_{j})\ln n(q_{j})+const$$

In a general case, entropy -based prior can be written as:(42)$$\ln P(\bar{f})=S=-\underset{k}{\sum }\beta_{k}\underset{j\in C_{k}}{\sum }{\bar{f}}_{j}\ln{\bar{f}}_{j},$$
where $\beta_{k}$ is a local regularization parameter. Substitution of the entropy prior (42) and condition probability (8) into (6) gives the Maximum a Posteriori estimation with entropy-based prior and local regularization (MAP-ENT-LOC):(43)$$\tilde{\bar{f}}=arg_{\bar{f}}\max\{-\underset{k}{\sum }\beta_{k}\underset{j\in C_{k}}{\sum }{\bar{f}}_{j}\ln{\bar{f}}_{j}+\underset{i}{\sum }[g_{i}\ln\underset{j}{\sum }a_{ij}{\bar{f}}_{j}-\underset{j}{\sum }a_{ij}{\bar{f}}_{j}-\ln g_{j}!].$$

## 3. Numerical Simulations

Numerical experiments were performed to study the local regularization method and to compare it with global regularization method. Figure 2a shows the two-dimensional (2D) clinical image in a torso axial cross section. One can see liver and stomach in the image. A liver lesion is shown as the bright area (hot spot) in the upper left part of the image. The image was obtained in SPECT with 111In-octreotide in Blokhin Russian Cancer Research Center. The appropriate 2D mathematical model was developed, as shown in Figure 2b. The tumor size in the model is smaller than the true tumor. Tumor activity (concentration of the radiopharmaceutical) was modeled 1.6 times higher than in the surrounding healthy liver tissues. In numerical simulations, the 2D model was sampled on the grid 128 × 128. Projection data were generated for 64 angular views over. The data acquisition model included the effects of non-uniform attenuation, collimator-detector response, and Poisson statistics [21]. Reconstruction of images from simulated data was performed by using the following four algorithms:OSEM—the standard non-regularized algorithm (9) applied on the most SPECT and PET systems;MAP-GIBBS—the Bayesian algorithm Maximum a Posteriori (MAP) with prior model based on the Gibbs distribution (14), global regularization;MAP-ENT—the MAP algorithm with prior model based on the entropy functional (34), global regularization; and,MAP-ENT-LOC—the MAP algorithm with prior model based on the entropy functional for open system (43), local regularization.

In numerical experiments, the behavior of these algorithms was studied. It should be noted that local regularization is a time consuming problem. In our simulation, the general reconstruction was performed by using the MAP-ENT with global regularization, and only for the selected organ (the liver) the MAP-ENT with local regularization was applied. The prior model was used in the form (42) with two regularization parameters: one for the healthy liver tissues and the second for the tumor. As it is assumed that we ‘do not know the location of the tumor in advance’, the second parameter was included in reconstruction algorithm in some iteration, provided that the activity in the voxel has exceeded the value of normal activity in healthy tissues. At that, we assume that the algorithm ‘knows or can estimate’ the level of activity in the healthy tissues.

## 4. Results and Discussions

The OSEM algorithm with eight subsets was applied in the first simulation study. Progress in images reconstructed after 1-st, 2-nd, and 5-th iterations is shown in Figure 3a–c. The profiles of activity along the line shown by arrow in Figure 2b were calculated and they are demonstrated in Figure 3d. OSEM is not a regularized algorithm; therefore, its behavior in iteration process is unstable and noisy artifacts dominate the images. Noise progresses more rapidly in the zones with lower statistics. In Figure 3a, one can see that after the first iteration, the image shows the presence of a tumor, but its activity, as is seen in Figure 3d, is underestimated and the borders are blurred. Noise peaks already have appeared on image at the second iteration. Therefore, the iterative process should have been stopped after the first iteration; however, image deterioration more rapidly progresses in the zones with lower count statistics. At a stopping point, a part of the image with low count statistics could already become noisy while another part of the image with high statistics have not reached its optimal resolution yet. This problem is especially important in oncology, because the noise in the low count zones can imitate false ‘hot spots’.

The MAP-GIBBS algorithm was used in the second simulation study. The appropriate value of the global regularization parameter β was empirically determined. Figure 4a–c show images that were reconstructed after 10-th, 15-th, and 20-th iterations. Figure 4d shows corresponding profiles of activity. Simulations were performed with different potential functions. In the presented pictures, the potential (15) was used. Rather minor differences were obtained between the images reconstructed by the MAP-GIBBS with potential functions (15) and (16). Analyzing the results presented in Figure 4, one can see that MAP-GIBBS algorithm blurs borders of the tumor. The underlying reason is that MAP-Gibbs algorithm produces the images that support local spatial correlations in the source function to be reconstructed.

The MAP-ENT algorithm was applied in the third simulation study. Figure 5a–c show images reconstructed after 10-th, 15-th, and 20-th iterations. Figure 5d shows the corresponding profiles of activity. As it was expected, in the absence of correlations in the MAP-ENT algorithm is superior to the MAP-GIBBS in identifying ‘hot spots’.

MAP-ENT-LOC algorithm was used in the fourth simulation study. Figure 6a–c show images that were reconstructed after 10-th, 15-th, and 20-th iterations. Figure 6d shows corresponding profiles of activity. It is seen that local regularization leads to improved sensitivity in tumor image in both the activity value and border identification.

In these simulations, the hot spot is a region of interest. Image quality is evaluated by using the profiles. The reconstructed tumor boundary and tumor activity are visually evaluated in comparison to the exact profile. The behavior of the MAP algorithms in an iterative process depends on the value of the regularization parameter. The parameter was empirically chosen. Further studies will be performed to derive a statistical criterion for choosing the optimal regularization parameter and stopping the iterative process.

## 5. Conclusions

In the Introduction, we suggested the hypothesis that disadvantages in images reconstructed by MAP algorithms are associated with global regularization. The new algorithm MAP-ENT-LOC with local regularization is developed. The suggested hypothesis was checked in computer simulations close to real clinical case. Four algorithms were compared: OSEM, MAP-GIBBS, MAP-ENT, and MAP-ENT-LOC. The comparative analysis has shown that local regularization leads to improved sensitivity in tumor image in comparison to global regularization in both activity value and border identification. Joint clinical and simulation studies are necessary for further development of this new advanced approach.

## Figures and Tables

**Figure 1 entropy-21-01108-f001:**
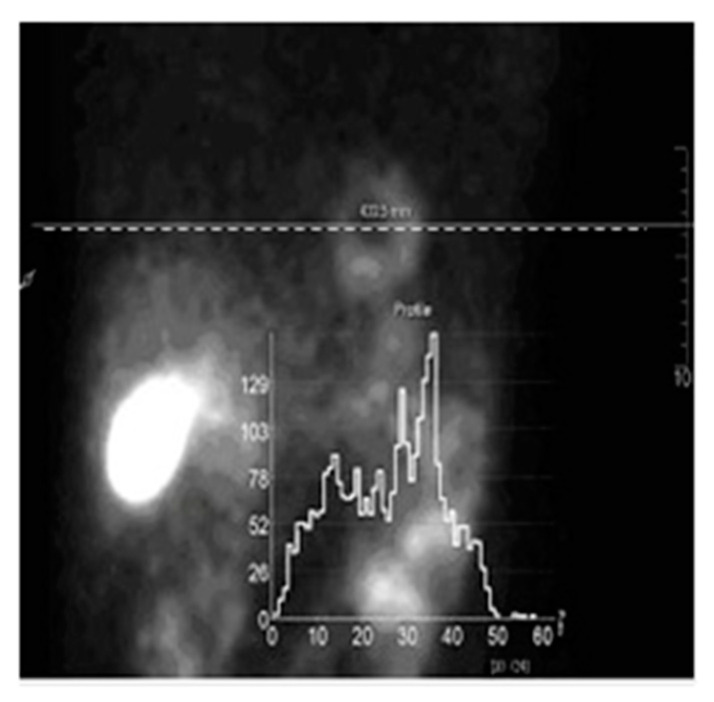
Single photon emission computer tomography (SPECT) myocardial perfusion imaging. Clinical LAO projection. The dashed white line indicates the position of the selected slice, the data for this slice are presented below. One can see that the data have stochastic properties. The clinical data were obtained using a Philips BrightView XCT SPECT/CT hybrid system in the National Medical Research Center of Cardiology (Moscow).

**Figure 2 entropy-21-01108-f002:**
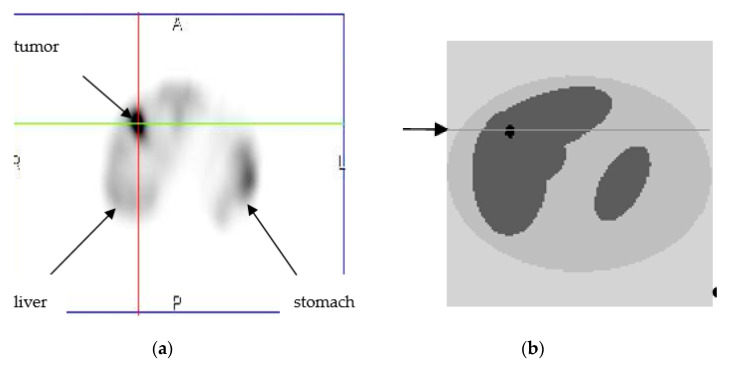
(**a**) Two-dimensional (2D) clinical liver image obtained with 111In-octreotide by using SPECT. The image is obtained in Blokhin Russian Cancer Research Center; and, (**b**) 2D mathematical model developed in the present paper.

**Figure 3 entropy-21-01108-f003:**
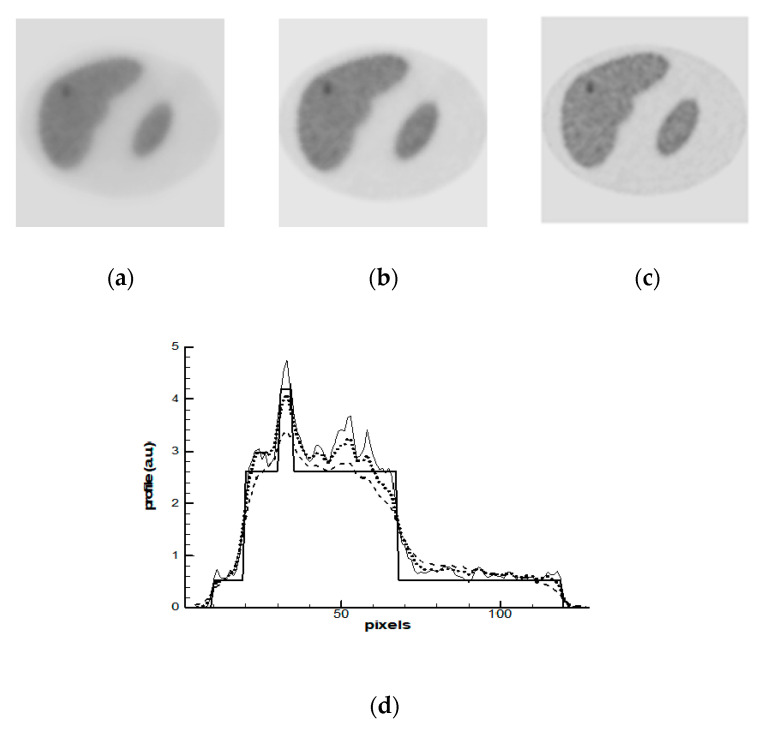
Ordered Subset Expectation Maximization (OSEM) reconstruction: (**a**–**c**) images reconstructed after 1-st (**a**), 2-nd (**b**), and 5-th (**c**) iterations; and, (**d**) profiles: the bold solid line—the exact profile (Figure 2b); the dashed line—the solution after 1-st iteration; the dotted line—the solution after 2-nd iteration and thin solid line is the solution after 5-th iterations. The profiles are demonstrated along the line shown by arrow in Figure 2b and Figure 3a.

**Figure 4 entropy-21-01108-f004:**
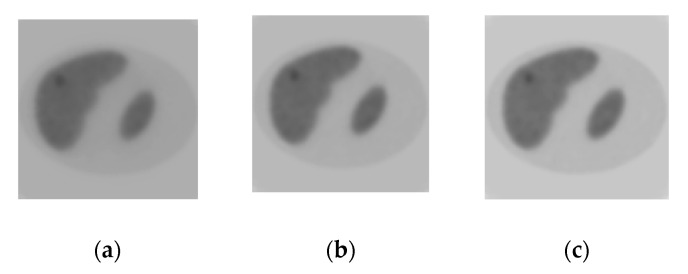
Bayesian method Maximum a Posteriori with Gibbs a priori probability (MAP-GIBBS) reconstruction: (**a**–**c**) images reconstructed after 10-th (**a**) and 15-th (**b**) and 20-th (**c**) iterations; and, (**d**) profiles: the bold solid line—the exact profile; the dotted line—the solution after 10-th iterations; the dashed line—the solution after 15-th iterations and thin solid line is the solution after 20-th iterations.

**Figure 5 entropy-21-01108-f005:**
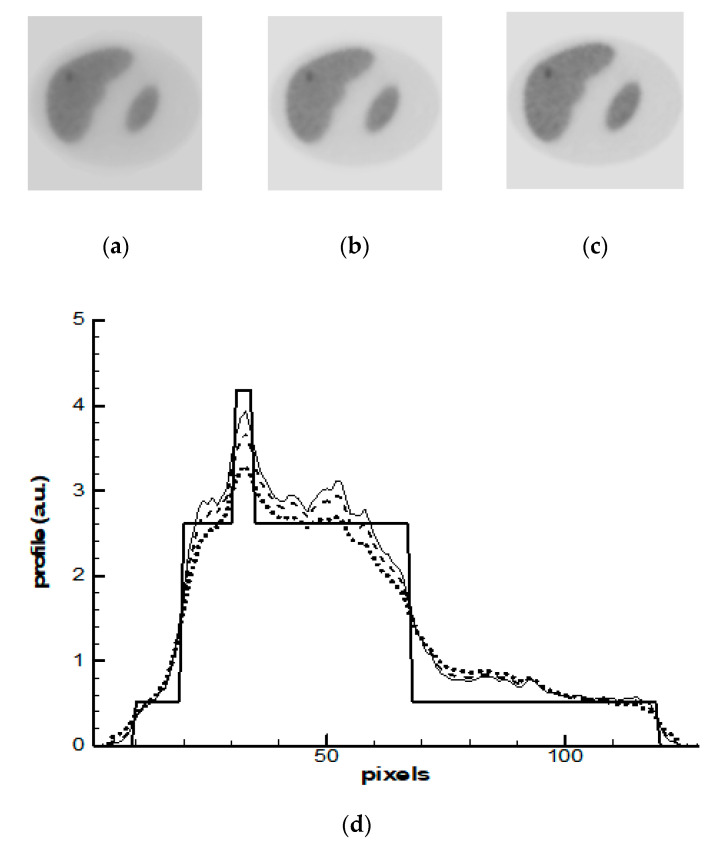
Bayesian method Maximum a Posteriori with entropy-based prior probability (MAP-ENT) reconstruction: (**a**–**c**) images reconstructed after 10-th (**a**), 15-th (**b**), and 20-th (**c**) iterations; and, (**d**) profiles: the bold solid line—the exact profile; the dotted line—the solution after 10-th iterations; the dashed line—the solution after 15-th iterations and thin solid line is the solution after 20-th iterations.

**Figure 6 entropy-21-01108-f006:**
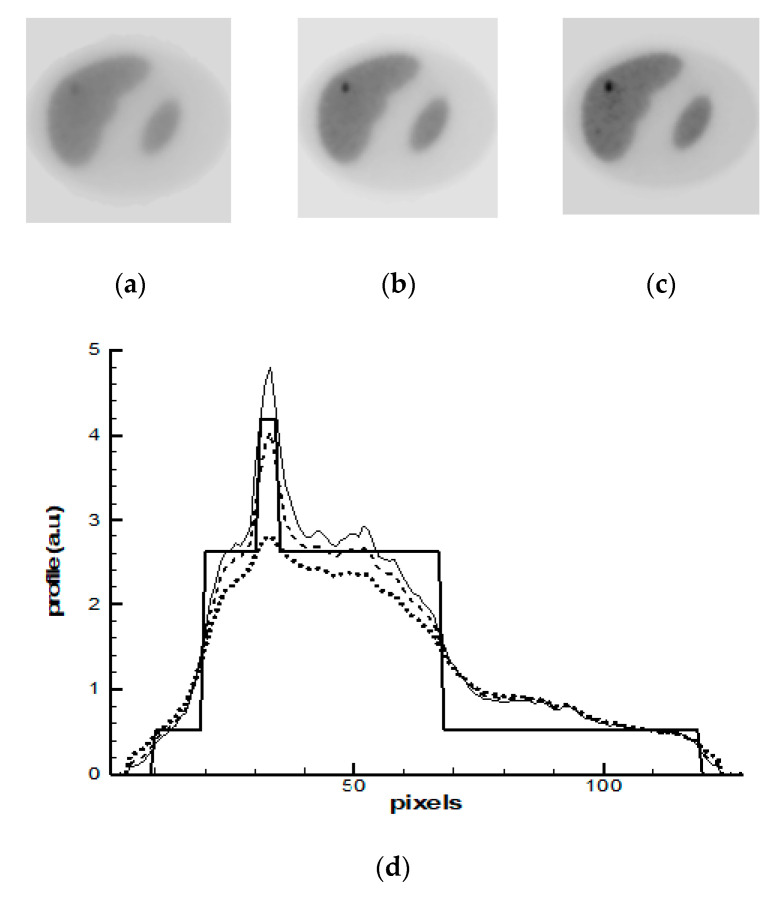
Maximum a Posteriori estimation with entropy-based prior and local regularization (MAP-ENT-LOC) reconstruction: (**a**–**c**) images reconstructed after 10-th (**a**), 15-th (**b**), and 20-th (**c**) iterations; and, (**d**) profiles: the bold solid line—the exact profile; the dotted line—the solution after 10-th iterations; the dashed line - the solution after 15-th iterations and thin solid line is the solution after 20-th iterations.

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
