# Peer review of "Bayesian Maximum-A-Posteriori Approach with Global and Local Regularization to Image Reconstruction Problem in Medical Emission Tomography"

_entropy, 2019, doi:10.3390/e21111108_

Round 1
Reviewer 1 Report
The paper "Bayesian maximum-a-posteriori approach with global and local regularization to image reconstruction problem in medical emission tomography" compares several image reconstruction methods. The paper examines the problems of relying on global regularization and demonstrates for an medical imaging problem improvements obtained from use of local regularization. Both theoretical explanations and experimental evidence are provided.
The paper is of high quality and clearly written; however, the English grammar needs to be corrected throughout the paper. I've indicated some of the grammar problems, but a more careful check is required.
One area where the technical content should be improved is the closing paragraph of section 2.3.1. A stronger statement can be made regarding the relationship of the potential function and the beta parameter to the physical model described in the next section.
More details on this and the grammatical issues are provided in the commented pdf which I've attached.

Author Response
First of all, thank you very much for your analysis of the manuscript.
The manuscript is revised substantially.
Vector-based description has been entered (lines 87-110).
Point1. One area where the technical content should be improved is the closing paragraph of section 2.3.1. A stronger statement can be made regarding the relationship of the potential function and the beta parameter to the physical model described in the next section.
Response1. The following text is written (lines 187-188)
‘The parameter is a constant and leads to global regularization in MAP-GIBBS algorithm (14).’
Point 2.More details on this and the grammatical issues are provided in the commented pdf which I've attached.
Response 2. The revisions have been performed in all the text in accordance with your comments.

Reviewer 2 Report
General Comments
In “Bayesian maximum-a-posteriori approach with global and local regularization to image reconstruction problem in medical emission tomography” the author proposes a maximum-a-posteriori (MAP) algorithm that uses “localized” regularization, as opposed to the more widely-used global regularization. In a few words, the author recognizes that different regions of a reconstructed object might require different trade-offs between a-priori information on the reconstruction and agreement with the measured data. Hence, the regularizing factor is split into two or more terms, each one with it is own regularization parameter (or set of parameters).
I have found no major flaws in the paper. However, I found many places where some improvement is needed to make the derivation clear. One thing I want to bring up right away (which may or may not be a big deal) is that the comparison between existing methods and the one proposed in the paper reports three reconstructed images (for example, after 10, 15, and 20 iterations) for each of the existing methods. On the other hand, for the proposed method, only one reconstruction (after 10 iterations) is reported. This seems a little bit unfair and I wonder what happens if the proposed algorithm is let run a little longer.
There are quite a few English mistakes or poor choices of words throughout the paper. I am confident these can be ironed down during editing once the technical aspects of the paper have been fully vetted.
Line-by-line Comments
Line 18: What does “more accurate” mean? The author should specify a figure-of-merit and use it to make comparison. It could be just difference between pixel value in the original object and the reconstructed object. Or it could also be something more meaningful and related to the ability of a radiologist to perform a correct diagnosis. Similarly, line 38 cites speaks about “higher quality” without defining what that means.
Line 51: What does “known” mean in “known paper”?
Line 97: Because A is mentioned to be a matrix, one must conclude that f is a vector. However, line 90 claims that f(q) is a function. Which one is correct?
Line 102: If g is a vector, how is the Poisson law defined on it? Shouldn’t the probability in Equation 3 given by the product of Poisson probabilities? Please clarify.
Line 111: If f is a function (see line 90) as opposed to a vector, how is the integral over f defined?
Line 115: I think “arg” is missing here.
Lines 210-230: Some reference is needed here.
Line 229: What is meant by “pulses”?
Line 232: Why has q become n in this equation? What is j?
Line 263: What is N?
Line 280: What does “default model” mean?
Line 287: I think f_j^n is missing on the right-hand-side of this expression.
Line 357: Something is wrong with the sentence that starts as “In these experiments …”
Author Response
First of all, thank you very much for your detailed review.
Point1.I have found no major flaws in the paper. However, I found many places where some improvement is needed to make the derivation clear. One thing I want to bring up right away (which may or may not be a big deal) is that the comparison between existing methods and the one proposed in the paper reports three reconstructed images (for example, after 10, 15, and 20 iterations) for each of the existing methods. On the other hand, for the proposed method, only one reconstruction (after 10 iterations) is reported. This seems a little bit unfair and I wonder what happens if the proposed algorithm is let run a little longer.
Response1. Such simulation studies were performed and, in the revised paper, Figure 6 shows the MAP-ENT-LOC images after the 10th, 15th and 20th iterations. The following text is added (lines 421-424):
‘The behavior of MAP algorithms in an iterative process depends on the value of the regularization parameter. The parameter was chosen empirically. Further studies will be performed to derive a statistical criterion for choosing the optimal regularization parameter and stopping the iterative process.’
Point 2. Line 18: What does “more accurate” mean? The author should specify a figure-of-merit and use it to make comparison. It could be just difference between pixel value in the original object and the reconstructed object. Or it could also be something more meaningful and related to the ability of a radiologist to perform a correct diagnosis. Similarly, line 38 cites speaks about “higher quality” without defining what that means.
Response 2. Quantitative evaluation of image quality is not a simple problem. By using models (phantoms) we can estimate the difference between pixel values in the phantom and the reconstructed object. However, in medicine, there is ‘the region of interest’ (ROI). Left ventricle is a ROI in cardiology, a hot spot is the ROI in oncology. In these cases, the quantitative evaluation of image quality can be performed as the difference between pixel values in the phantom and the reconstructed object in the ROI.
The following text is added (Lines 421-423):
‘In these simulations, the hot spot is a region of interest. Image quality is evaluated by using the profiles. The reconstructed tumor boundary and tumor activity are evaluated visually in comparison to the exact profile.’
Point3. Line 51: What does “known” mean in “known paper”?
Response3. This is corrected.
Point 4. Line 97: Because A is mentioned to be a matrix, one must conclude that f is a vector. However, line 90 claims that f(q) is a function. Which one is correct?
Response 4. This is revised (lines 87-110). Vector-based description has been entered.
Point 5. Line 102: If g is a vector, how is the Poisson law defined on it? Shouldn’t the probability in Equation 3 given by the product of Poisson probabilities? Please clarify.
Response 5. This is also revised.
Point 6. Line 111: If f is a function (see line 90) as opposed to a vector, how is the integral over f defined?
Response 6. This is corrected. Now, the Bayesian Maximum a Posteriori probability is written in vector form.
Point 7. Line 115: I think “arg” is missing here.
Response 7. This is corrected (line114).
Point 8. Lines 210-230: Some reference is needed here.
Response 8. The reference [26] is included (lines 220 and 222).
Point 9. Line 229: What is meant by “pulses”?
Response 9. This is corrected (Lines 229-230):
‘Substituting (18) into (17) and integrating the Gibbs distribution (17) on momenta one obtains:’
Point 10. Line 232: Why has q become n in this equation? What is j?
Response 10. This is corrected. Eq.(23) is written for vector of particles n^j.
The explanation is given (lines 204-206):
‘For simplicity, we assume that the volume of each voxel is unit and the concentration determines the number of particles in the voxel .’
Point 11. Line 263: What is N?
Response 11. This is corrected. This is N0.
Point 12. Line 280: What does “default model” mean?
Response 12. We are going to study this form in our future work (lines 282-283):
“In our next work, we are going to study and compare the forms (30) and (31) in application to nuclear medicine. In the present work, the entropy form (30) was used.’
Possible prior information about the structure of the solution could be encoded in the default model.
Point 13. Line 287: I think f_j^n is missing on the right-hand-side of this expression.
Response 13. Not missing. For explanation, we added the text (lines 291-293)
‘In our calculations, the multiplicative form of the MAP-ENT algorithm with was used:
,
Point 14. Line 357: Something is wrong with the sentence that starts as “In these experiments …”
Response 14. This sentence has been removed.

Reviewer 3 Report
The manuscript is well-written. I would like to recommend it to be accepted in current form.
Author Response
Thank you very much for your review and support.
Round 2
Reviewer 2 Report
The author implemented most, if not all, my suggestions and, as a result, I feel the paper is suitable for publication. I just have a few comments, which I report below.
Line 38: I still feel uncomfortable with statement about image quality without a proper definition and/or image quality study. In this case, I feel that stating "... produces images with higher quality ..." is a too strong statement for the Introduction. I propose the following rewording: "... produces images deemed of higher quality ...".
Equation 2: Because gi are noisy data, there might not be any non-negative vector f that satisfy this equation. What is actually true is that the mean of the gi satisfy this equation. One way to see this (and to fix the notation) is to interpret aij as the probability of a photon emitted from voxel j is detected in detector pixel i. Because fj is a Poisson random variable, aijfj is also a Poisson random variable (this follows from the binomial selection theorem, which states that the random selection of a Poisson random variable X with probability p is also a Poisson random variable with mean p * mean(X)). Moreover, all the Poisson random variables aijfj are independent. Therefore, their sum is a Poisson random variable. Hence, the sum in equation 2 is the mean of gi. In other words, there are two different stochastic mechanisms that take place in the production of gi counts: (1) generation of fj counts from voxel j; and (2) acceptance of these emitted counts with probability aij each. I suggest the author to either place a bar over gi or to add a zero-mean noise term ni to the left-hand-side.
Equation 23: There is an extra K factor in front of exp[...].
Lines 393 and 407: It should be Figure 2(a).
Author Response
Reply to the Ref1 (round2)
Thank you very much for your detailed review.The reply is in attached file.
